

# On the impact of masking and blocking hypotheses for measuring the efficacy of new tuberculosis vaccines

Sergio Arregui[1,2], Joaquín Sanz[1,3,4], Dessislava Marinova[5,6], Carlos Martín[5,6,7] and Yamir Moreno[1,2,8]

[1] Institute for Biocomputation and Physics of Complex Systems (BIFI), University of Zaragoza, Zaragoza, Spain
[2] Department of Theoretical Physics, University of Zaragoza, Zaragoza, Spain
[3] Sainte-Justine Hospital Research Centre, Montreal, Quebec, Canada
[4] Department of Pediatrics, University of Montreal, Montreal, Quebec, Canada
[5] Department of Microbiology, Faculty of Medicine, University of Zaragoza, Zaragoza, Spain
[6] CIBER Enfermedades Respiratorias, Instituto de Salud Carlos III, Madrid, Spain
[7] Service of Microbiology, Miguel Servet Hospital, Zaragoza, Aragón, Spain
[8] Complex Networks and Systems Lagrange Lab, Institute for Scientific Interchange, Turin, Italy

Corresponding author
Sergio Arregui, sergioarregui.sa@gmail.com

## ABSTRACT

Over the past 60 years, the *Mycobacterium bovis* bacille Calmette–Guérin (BCG) has been used worldwide to prevent tuberculosis (TB). However, BCG has shown a very variable efficacy in different trials, offering a wide range of protection in adults against pulmonary TB. One of the most accepted hypotheses to explain these inconsistencies points to the existence of a pre-existing immune response to antigens that are common to environmental sources of mycobacterial antigens and *Mycobacterium tuberculosis*. Specifically, two different mechanisms have been hypothesized to explain this phenomenon: the masking and the blocking effects. According to masking hypothesis, previous sensitization confers some level of protection against TB that masks vaccine's effects. In turn, the blocking hypothesis postulates that previous immune response prevents vaccine taking of a new TB vaccine. In this work we introduce a series of models to discriminate between masking and blocking mechanisms and address their relative likelihood. We apply our methodology to the data reported by BCG-REVAC clinical trials, which were specifically designed for studying BCG efficacy variability. Our results yield estimates that are consistent with high levels of blocking (41% in Manaus -95% CI [14–68]- and 96% in Salvador -95% CI [52–100]-). Moreover, we also show that masking does not play any relevant role in modifying vaccine's efficacy either alone or in addition to blocking. The quantification of these effects around a plausible model constitutes a relevant step towards impact evaluation of novel anti-tuberculosis vaccines, which are susceptible of being affected by similar effects, especially if applied on individuals previously exposed to mycobacterial antigens.

## INTRODUCTION

Despite all the efforts in the fight against TB accomplished during the last decades, the disease still remains a major cause of morbidity and mortality worldwide, being responsible for a million and a half deaths per year all around the world (*World Health Organization Global Tuberculosis Report, 2014*). The increasing emergence of multi drug and extremely drug-resistant strains (*Dye, Williams & Williams, 2000*) or the association between TB and VIH (*Boily, Lowndes & Alary, 2002*; *Korenromp et al., 2003*) constitute serious epidemiological threats that evidence the necessity of further public health measures and pharmacological resources against the disease.

Among all the possible epidemiological interventions that could contribute to the desired goal of TB eradication, the introduction of a novel preventive vaccine is currently thought to be able to offer the highest and most immediate impact on disease burden reduction. The efficacy of the current TB vaccine BCG is consistent in protecting infants, especially from the most severe forms of meningeal and miliary TB (*Mangtani et al., 2014*) but is limited against pulmonary forms of the disease responsible for transmission fueling the growing epidemic worldwide. Accordingly, nowadays there exist more than fifteen different research teams worldwide developing as many novel experimental vaccine candidates designed as revaccination (boosting) strategies in BCG vaccinated individuals (adolescents or adults) or as a BCG replacement strategies at birth (*Marinova et al., 2013*).

BCG fails to provide consistent protection to the pulmonary forms of the disease, especially in adults (*Rodrigues, Diwan & Wheeler, 1993*), who are the main contributors of overall disease spreading. Consequently, an accurate evaluation of the BCG impact under different conditions—population susceptibility, geography, environmental exposure, etc.—is essential. Such an evaluation will allow the assessing of the efficiency of BCG as a reference vaccine and, at the same time, will provide new guidelines and methodological tools to better evaluate the potential efficacy of the newly developed TB vaccines. The highly variable and apparently inconsistent results obtained in BCG's efficacy tests and meta-analysis have been subject of intense scientific controversy (*Mangtani et al., 2014*; *Barreto et al., 2014*), and the use of BCG during the 20th century has been largely argued (*Fine & Rodrigues, 1990*; *Bloom & Fine, 1994*).

The hypothesized causes underlying the observed variability of BCG efficacy in different settings include differences between the BCG strains (*Oettinger et al., 1999*), genetic, epi-genetic or socio-economical differences between populations, study quality, parasitic co-infections, etc (*Marinova et al., 2013*). In addition, multi-variate meta-analysis of BCG efficacy determination studies consistently determine that latitude is a variable showing a most prominent correlation with BCG performance (*Fine & Rodrigues, 1990*; *Fine, 1995*; *Brewer, 2000*; *Zodpey & Shrikhande, 2007*), pointing to the existence of latitude-driven mechanisms influencing it, rather than other possible explanations related, for example, to the ethnicity of the tested populations (*Fine et al., 1999*). Among these possible mechanisms, the hypothesis that agglutinates a greater consensus points to the existence of a complex, latitude-dependent immunological process of environmental sensitization (ES) to mycobacterial antigens which might interfere with the observed

action of BCG vaccine in different ways. The hypothesis of ES being the source of BCG efficacy variability has been backed up by different epidemiological observations (*Hart & Sutherland, 1977*; *Miceli et al., 1988*; *Al-Kassimi et al., 1995*; *Mangtani et al., 2014*).

ES is thought to have its origin in the exposure of individuals either to non tuberculous mycobacteriae (NTM)—whose antigenic similarity to MTB (*Chaparas, Maloney & Hedrick, 1970*; *Harboe et al., 1979*) is able to cause cross reactivity in the human immune system (*Black et al., 2001*; *Weir et al., 2003*)—or to the reservoir of latent infection of MTB itself (and other closely related bacteria within the MTB-complex). Additional sources of sensitization have been postulated, like certain parasitic infections (*Ferreira et al., 2002*). The diversity among the different putative sources of ES is notorious, the relationship between their prevalence and latitude is not homogeneous, and their levels of cross reactivity are variable as well. This situation portraits a complex landscape that makes specially ventured to attribute the geographical patterns of BCG efficacy variation to a single factor, as it could be a global increase in NTM prevalence levels next to the equator (*Black et al., 2001*; *Floyd et al., 2002*; *Weir et al., 2003*), which has been demonstrated to be inaccurate for some species (*Hoefsloot et al., 2013*). Even though, it seems clear that overall levels of ES increase both with closeness to equator and subjects' age at the time of vaccination.

Two different mechanisms have been theorized on how this exposition to environmental antigens would affect the response of the host to a vaccine like BCG (*Andersen & Doherty, 2005*). The masking hypothesis postulates that ES confers a significant protection against TB in such a way that a vaccine can barely offer an additional level of protection (*Palmer & Long, 1966*; *Andersen & Doherty, 2005*). As an alternative hypothesis, it has been suggested that ES prior to vaccination may trigger an immune response capable of blocking the assimilation of the vaccine by the host, either if it's a live-attenuated vaccine or if it's a booster. This is known as the blocking hypothesis (*Brandt et al., 2002*; *Andersen & Doherty, 2005*). These two effects have the potential to explain, to a large extent, the variability observed in the trials performed, that is, both the dependence of BCG efficacy on age at the time of vaccination—as an individual gets older its exposition to mycobacteriae increases—and its geographical variations. Finally, it is worth highlighting that masking and blocking do not exclude each other: in a scenario in which both mechanisms take place, ES would contribute at the same time to reduce disease risk of non-vaccinated individuals, and to impair vaccine assimilation of immunized ones.

Several studies have tackled the problem of evaluating ES impact on BCG vaccine efficacy from different angles. Researches in animal models have shown that environmental mycobacteria strains can interfere with BCG vaccination and with susceptibility to M. tuberculosis infection (*Hernandez-Pando et al., 1997*; *Demangel et al., 2005*; *Young et al., 2007*). The influence of the effects of Masking and Blocking on measurements of efficacy has been also studied from a theoretical point of view (*Fine & Vynnycky, 1998*; *Mantilla-Beniers & Gomes, 2009*), even though none of these works allows a quantification of masking and blocking effects on vaccine's efficacy levels measured on clinical trials performed on humans.

BCG-REVAC trials were designed to discriminate these two effects on BCG performance when applied on individuals of dissimilar ages in the Brazilian cities of Salvador and Manaus (*Barreto et al., 2002*; *Rodrigues et al., 2005*; *Barreto et al., 2014*). In particular, three types of trials were conducted in the study, measuring the efficacy of as many vaccination strategies in each city: newborn vaccination, school-age vaccination and school-age revaccination. The rationale behind the election of such a design is twofold. On the one hand, replicating the experiment design in two cities of the same country located at considerably different latitudes, renders reasonable the assumption that the main source of variability at the efficacies observed is due to different levels of ES, since virtually any other plausible source of variation (i.e., vaccine preparation, strain or application protocol, ethnic diversity etc.) are absent or controlled for across the study. On the other hand, the trials design allows discriminating between blocking and masking effects, since the differences across cities of the efficacies observed for each type of trial are expected to vary depending on what of the two effects is dominant.

After the analysis of BCG-REVAC trials, *Barreto et al. (2014)* observed that the efficacy of the vaccine, when applied to newborns and measured later in life did not show a strong geographic variation, which suggests that spontaneous protection related to masking should play a residual role, if any. On the contrary, when BCG was applied at school age (*Barreto et al., 2014*), either the first time, or as a second dose, vaccine efficacy observed was in both cases lower in Manaus than in Salvador; which in principle would be compatible with the blocking hypothesis if vaccine assimilation were more efficient in Salvador as a consequence of lower levels of ES bound to its larger distance to the equator. However, even if the design of BCG-REVAC trials allowed to qualitatively asses the greater relevance of the blocking mechanism as compared to masking, no actual quantification of these two effects and their relative role has been provided up to now. In this sense, after the work by the BCG-REVAC consortium, several questions remain unanswered, as we do not know (1) what is the relative likelihood of both hypothetical mechanisms when trying to explain the observed results of the trial, (2) how much predictive power would a full model containing both effects gain with respect to single effects scenarios (masking or blocking alone) (3) whether significantly different combinations of masking and blocking strengths could be similarly compatible with the observations derived from the trials, and very relevantly, (4) what are the intensities of blocking and masking effects, and their confidence intervals, yielding a most significant agreement with the data.

In this paper, we introduce a family of mathematical models to interpret the results from BCG-REVAC trial under the light of masking and blocking effects, in order to contribute to answering the aforementioned questions within the limitations imposed by the reduced statistical power derived from the reduced number of trials studied. By confronting our model against the results of the BCG-REVAC studies, we are able to measure extent to which these effects are sufficient to explain the efficacies measured (*Barreto et al., 2014*). Furthermore, we quantify the specific masking, blocking and immunity waning effects yielding best-fitted estimates for the efficacies measured. To this end, we compared the likelihoods of three different modeling scenarios: a first model in which both effects concurrently take place, a second model only considering blocking

and a third one containing only masking. Translating the trial results into quantitative estimations of blocking and masking strengths constitutes a relevant step towards a deeper knowledge on how BCG efficacy depends on individuals' age and geographical areas. Similarly, it would provide a quantitative reference for the plausible ranges of blocking and masking levels that other TB vaccines might eventually suffer as well, which could be especially relevant in the context of impact evaluation of novel vaccines. Up to our knowledge, the BCG-REVAC studies are the only set of trials specifically designed to discriminate the effects of Masking and Blocking. Even though our work is restricted to this particular setting, our framework could also be extended to the interpretation of future trials and impact evaluation of future vaccines.

## METHODS

### Data analyzed: the Brazilian BCG-REVAC clinical trials

BCG-REVAC consisted of a set of cluster-randomized trials involving more than 200,000 school-aged children in the Brazilian cities of Manaus and Salvador, whose principal aim was evaluating the effectiveness of BCG under different vaccination protocols.

The enrolled population of the study consisted of non-infected school children between 7 and 14 years old at the moment of randomization. Within this population, individuals presenting a positive BCG scar are separated from the rest, distinguishing, this way, the enrolled individuals who were vaccinated at birth from those who were not. Each group is then split into an intervention and a control group; individuals in the intervention group were vaccinated within the context of the trial. Summing up, there are 4 cohorts in each city: non-vaccinated (1), vaccinated after birth (2), firstly vaccinated at school age during the trial (3), and revaccinated, after a first dose applied after birth, in the trial too (4). Upon such classification of enrolled individuals in cohorts, the effectiveness of BCG vaccination strategies was measured by comparing the TB incidence rate within an end-point associated to active disease in the four cohorts, according to three different types of trials: Trial I: BCG at birth vs. no intervention (cohort 2 vs. 1). Trial II: BCG first dose at school age vs. no intervention (cohort 3 vs. 1). Trial III: revaccination at school age vs. first dose at birth only (cohort 4 vs. 2).

### A model to describe BCG efficacy variation: masking, blocking and immunity waning

The six clinical trials conducted within the framework of BCG-REVAC study-three types of trials per two cities- output efficacies that span from 1% to 40% protection (see Fig. 1, red continuous lines). In order to explain this variability, we propose a model according to which the different protection levels found in each of the four cohorts in the study, schematically shown in Fig. 2, result from the interplay between the intrinsic vaccine efficacy, its temporal waning patterns, masking and blocking effects. These three mechanisms of vaccine protection shifts are ultimately responsible for vaccine's performance variation, either in space or in time.

First of all, in absence of masking or blocking, a naive vaccinated individual will receive a protection level, right after vaccination, that we call $e(0)$. As time after vaccination goes

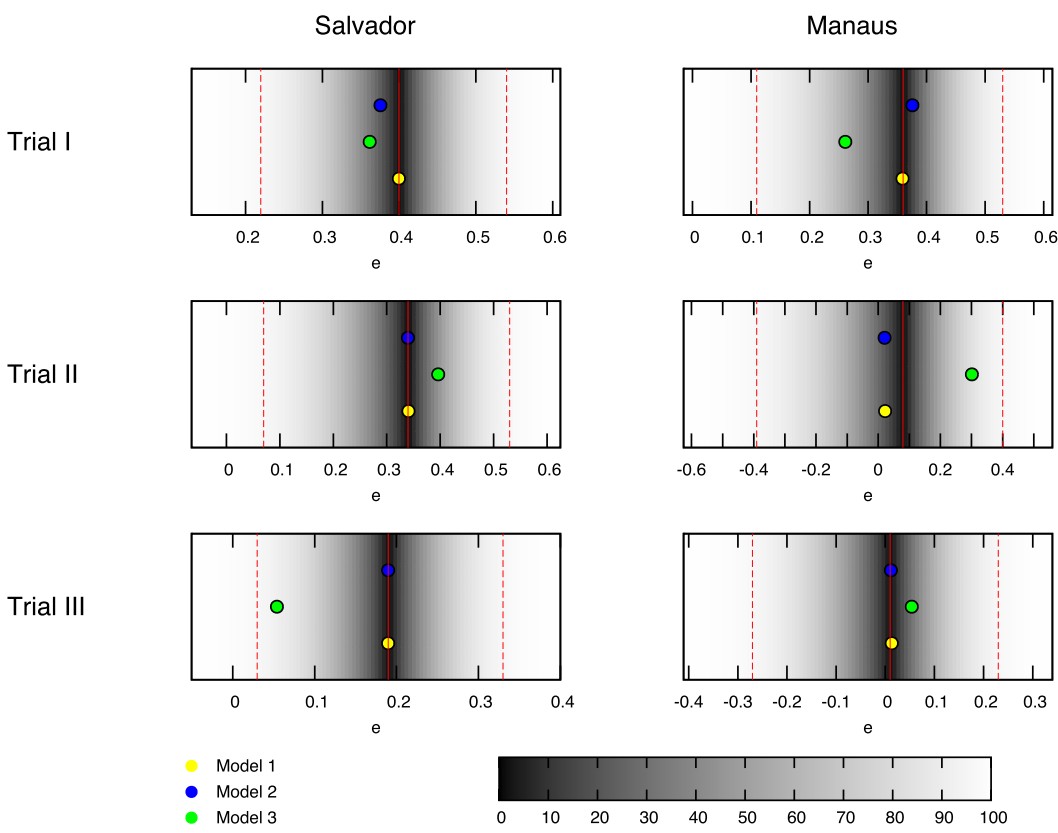

**Figure 1 Best fit estimates for each trial by models 1, 2 and 3 (yellow, blue and green dots, respectively) for the trials conducted in the BCG-REVAC study.** The colormap represents the probability of obtaining a less extreme value of the efficacy, according to the distributions considered. The probability of zero marks the central estimate (red, continuous line) while the dashed red lines mark the 95% CI reported by *Barreto et al. (2014)*.

by, this protection level will wane up to $e(t) < e(0)$, generally speaking. This implies that, if we deal with a population in which the incidence rate of new TB cases per unit time is equal to $x$; $t$ years after vaccination, this rate is modified to $(1 - e(t))x$, provided that no additional effects take place. Taking that into account, a protective vaccine will have positive efficacy values $e(t) \in (0, 1]$, being also possible for a (failed) vaccine to have a negative efficacy if it augments the disease risk among vaccinated individuals instead of reducing it. In our model, the time waning patterns of the intrinsic vaccine efficacy do not depend on the geographical area, but just on time since vaccination, which approximately is, in average, 4.5 years for school age vaccination (cohort 3) and 16 years for newborn vaccination (cohort 2), which implies the consideration of two intrinsic efficacy parameters: $e(4.5)$ and $e(16)$.

Besides vaccination, ES can also support protection against disease through the masking mechanism. The masking level, denoted by $m$, is a protection parameter formally equivalent to the intrinsic vaccine efficacy (thus verifying $m \in (0, 1]$ for a protective effect, and negative otherwise), whose effects are suffered by initially naive, non-vaccinated individuals subject to ES. Thus, in principle, the longer the time an individual has been
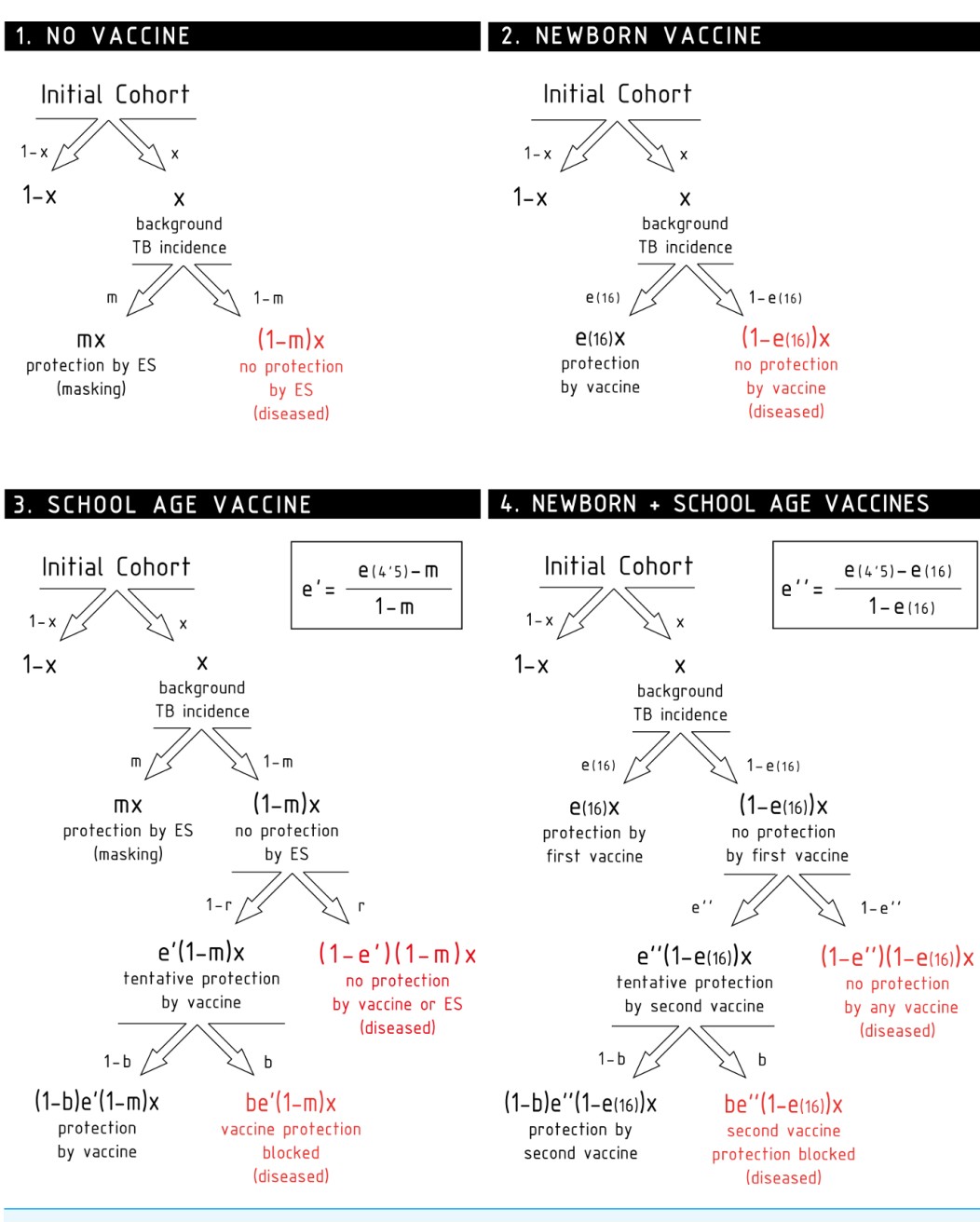

**Figure 2** Scheme of the different contributions to the disease risk for each cohort.

exposed to ES—i.e., the older the individual is at the moment of observation—the higher is the masking-related protection she might show. Masking is also a geography-dependent effect, since it depends on ES, which forces us to consider two masking parameters: $m^M$ for Manaus and $m^S$ for Salvador. The dependence of these parameters on age cannot be resolved, since all the cohorts analyzed in the study have approximately the same age.

Additionally, if $e(t)$ describes the protection provided by the vaccine to a naive individual in absence of masking or blocking $t$ years after vaccination, we also need to describe how this protection is modified if the vaccine is applied to non-naive subjects.

If an individual's immune system has been stimulated prior to vaccination (either by masking like in cohort 3, or by a previous vaccine, like in cohort 4 before the second dose), and consequently she is partially protected against the disease, it is unrealistic to assume that the full effect of the new dose is additive (*Andersen & Doherty, 2005*). Instead of that, our model considers that a vaccine dose applied on a previously protected individual will contribute, at most, up to resetting the initial protection levels $e(0)$, provided that no blocking of the vaccine takes place. In cohort 3, this implies that, right after the school age vaccination, if the vaccine is not blocked ($b = 0$, see below), it will have a protective effect $e'$ that will be concurrent with the masking protection $m$ so as to reduce the disease risk from $[1-m]x$ to $[1-e'][1-m]x$. Our estimation of $e'$ comes from assuming that such disease risk must equate what we would observe if a vaccine of full efficacy were applied on naive individuals, and observed 4.5 years later:

$$[1-e'][1-m]x = [1-e(4.5)]x \rightarrow e' = \frac{e(4.5)-m}{1-m}. \qquad (1)$$

Similarly, the school-age dose at cohort 4, will add to the protection provided by the newborn dose $e(16)$, diminishing the disease risk from $[1-e(16)]x$ to $[1-e''][1-e(16)]x$. To estimate $e''$, we assume that, if the second vaccine is not blocked, the disease risk achieved by both vaccines together $[1-e(16)][1-e'']x$ is equivalent to the disease risk reached by the same vaccine, if applied on unprotected individuals, 4.5 years after vaccination:

$$[1-e(16)][1-e'']x = [1-e(4.5)]x \rightarrow e'' = \frac{e(4.5)-e(16)}{1-e(16)}. \qquad (2)$$

Finally, vaccine intrinsic efficacy can be blocked by prior ES; an effect that we model through the blocking probability $b \in [0,1]$, where $b = 0$ means that no blocking appears, while $b = 1$ stands for a totally blocked vaccine, meaning that vaccinated individuals would only have the protection level that they already had before vaccination. Blocking is also a geography-dependent factor, since it is considered a consequence of ES as well, which forces us to distinguish $b^M$ and $b^S$ for Manaus and Salvador, respectively. Unlike masking, blocking does not depend on the age of the individuals at the moment of observation, but on their age at the moment of vaccination. In this case we study cohorts vaccinated at two moments in life—at birth and at the beginning of the trials—being the first of these cases (the newborn vaccination) considered blocking-free, as it is assumed that when the vaccine is applied immediately after birth, there is no place for prior ES.

Taking all these effects into account, we are left with a set of six independent parameters $\vec{P} = \{e(4.5), e(16), m^M, b^M, m^S, b^S\}$ to describe the variability observed in the trials, either temporal or geographical, under the light of blocking and masking effects, concurrently. The temporal trends of the level of protection of each cohort are schematically shown in Fig. 3.

In the following, we will refer to this full model as model 1. In Fig. 2, we represent the variations on the disease rates provoked by each effect that takes place in each cohort according to model 1. Summing all the possible *contributions* to the development of active

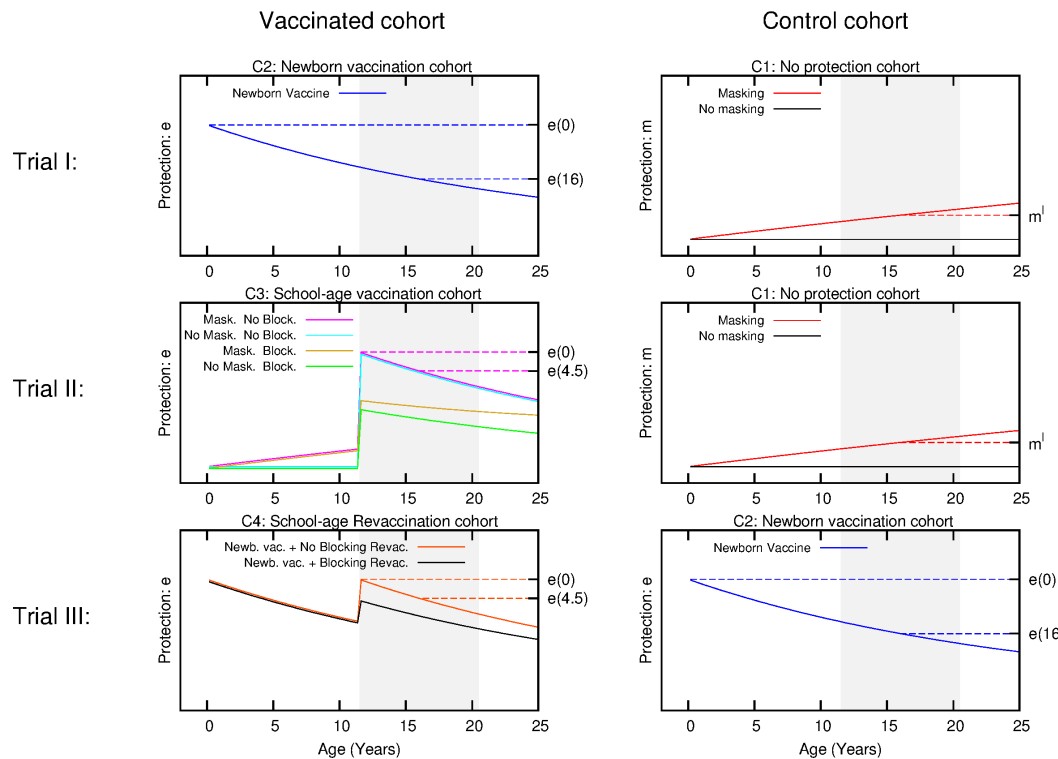

**Figure 3** **Scheme for the temporal evolution of the level of protection for the cohorts of the three types of trials considered in the work, according to the different vaccination strategies and ES mechanisms.** Trial I: the control group is cohort one that corresponds to non-vaccinated individuals. In this cohort, a level of protection above zero can only be due to masking, which is an increasing function with age. In turn, the intervention group corresponds to cohort 2 that is the newborn vaccination group: individuals are vaccinated right after birth, which provides a protection that overcomes any possible masking effect, cannot be blocked by ES and wanes with time. Trial II: The vaccinated cohort is cohort 3, firstly immunized at school age. In this cohort individuals might be protected by masking before the vaccine is applied. Then, at the moment of vaccination, if not blocked, the vaccine will overcome masking protection up to the initial value $e(0)$, which then will wane. Finally, if blocking takes place the protection provided by the vaccine will be reduced. The control cohort in this case is cohort 1 again. Trial III: Intervention group corresponds to cohort 4, joined by individuals firstly vaccinated at birth, and revaccinated at school age. At variance to the first dose, which cannot be blocked, the second dose might be blocked by ES or not, in which it will reset the initial protection levels provided by the vaccine. The control group for this trial is cohort 2, that corresponds to individuals only vaccinated at birth. The grey shaded area represents the age window of the individuals enrolled in the study.

disease for each cohort, we derive the general disease rates characterizing each cohort of one city as follows:

$$d_1^l = (1 - m^l)x$$
$$d_2^l = [1 - e(16)]x$$
$$d_3^l = (1 - b^l)[1 - e(4.5)]x + b^l(1 - m^l)x$$
$$d_4^l = (1 - b^l)[1 - e(4.5)]x + b^l(1 - e(16))x$$

(3)

where the superscript indicates location, and $x$ the incidence rate observed in the population.

From (3), it is immediate to derive the expressions for the observed efficacies $\bar{e}$ of each trial according to model 1, which read as:

$$M_1 : \begin{cases} \bar{e}_I^l = 1 - d_2^l/d_1^l = \dfrac{e(16) - m^l}{1 - m^l} \\[2mm] \bar{e}_{II}^l = 1 - d_3^l/d_1^l = \dfrac{e(4.5) - m^l}{1 - m^l}(1 - b^l) \\[2mm] \bar{e}_{III}^l = 1 - d_4^l/d_2^l = \dfrac{e(4.5) - e(16)}{1 - e(16)}(1 - b^l) \cdot \end{cases} \tag{4}$$

The system of Eq. 4 represents a full model for the vaccine efficacies observed during BCG-REVAC trials, which is based on the assumption that the sources of geographical variability for BCG's performance are both masking and blocking effects. From the full model, two reduced versions can be conceived: a masking-free model (model 2 in the following) in which $m^M = m^S = 0$, and a blocking free model in which $b^M = b^S = 0$ (model 3). The efficacies associated to each trial, for models 2 and 3 straightforwardly read as follows:

$$M_2 : \begin{cases} \bar{e}_I^l = e(16) \\[2mm] \bar{e}_{II}^l = e(4.5)(1 - b^l) \\[2mm] \bar{e}_{III}^l = \dfrac{e(4.5) - e(16)}{1 - e(16)}(1 - b^l) \end{cases} \tag{5}$$

$$M_3 : \begin{cases} \bar{e}_I^l = \dfrac{e(16) - m^l}{1 - m^l} \\[2mm] \bar{e}_{II}^l = \dfrac{e(4.5) - m^l}{1 - m^l} \\[2mm] \bar{e}_{III}^l = \dfrac{e(4.5) - e(16)}{1 - e(16)} \cdot \end{cases} \tag{6}$$

By considering these three models, our approach allows quantifying and comparing the plausibility of blocking and masking hypotheses to potentially explain the variation in BCG efficacy trials observed in the controlled setup conceived in the BCG-REVAC trials, taking into account the non-linearities associated to each mechanism, which play a central role in the derivation of Eqs. (4)–(6).

## Models solution: parameters estimation and confidence intervals

In order to identify the set or sets of parameters yielding a best fit for the efficacies observed in BCG-REVAC trials, we compare the model prediction associated to any parameter set $\vec{P}$ to a set of empirical probability distributions derived from BCG-REVAC data. From each of the confidence intervals reported in *Barreto et al., (2014)* we build a two-piece normal distribution (*Wallis, 2014*) for each trial reported, centered in the reported values $[\bar{e}_i^l]_{\text{BCG-REVAC}}$ (for location $l \in \{\text{Manaus, Salvador}\}$ and trial $i \in \{I, II, III\}$), and with asymmetric variances $[\sigma_i^l]_{\text{BCG-REVAC}}^{\pm}$ equal to one half the radius of the confidence intervals reported in *Barreto et al., (2014)*, so preserving the confidence levels of the intervals reported (see Fig. 1).

Once the empirical distributions have been defined, for each possible set of parameters and for each of the six trials we define the $Z$-score associated to the model prediction as:

$$Z_i^l(\vec{P}) = \left| \frac{\left[\bar{e}_i^l(\vec{P})\right]_{\text{mod}} - \left[\bar{e}_i^l\right]_{\text{BCG-REVAC}}}{\left[\sigma_i^l\right]_{\text{BCG-REVAC}}^{\pm}} \right| \qquad (7)$$

where $\left[\sigma_i^l\right]_{\text{BCG-REVAC}}^{\pm}$ will take each of its two possible values depending on the sign of $([\bar{e}_i^l(\vec{P})]_{\text{mod}} - [\bar{e}_i^l]_{\text{BCG-REVAC}})$. From $Z_i^l(\vec{P})$, we define the corresponding $p$-values $p_i^l[Z_i^l(\vec{P})]$ as the probability of the empirical distributions reproducing BCG-REVAC data to have a $Z$-score $\tilde{Z}$ so that $|\tilde{Z}| > |Z_i^l(\vec{P})|$. This allows us to define the following likelihood function:

$$L(\vec{P}) = \prod_{l,i} p_i^l[Z_i^l(\vec{P})] \qquad (8)$$

to maximize so as to identify the model's parameters $\vec{P}^*$ more likely to yield the BCG-REVAC results. The global landscape of $L(\vec{P})$ is explored using a hill-climbing algorithm designed to identify all possible local maxima in the space of parameters. Finally, a Levemberg-Marquardt algorithm is used to find a more accurate value of the global maximum, if the latter is unique.

In order to estimate the confidence interval associated to our model estimation, the following numerical procedure is performed. First, and starting from the maximum likelihood estimate $\vec{P}^*$, we move on each parameters' axis until a value of $L = 0.05$ is reached in each case. We call this increment $A_j$ ($j \in [1,6]$) (see Fig. 4). These values are not symmetrical, again, and so we distinguish between $A_j^+$ and $A_j^-$. Using these asymmetric widths, we construct a two-piece normal distribution for every parameter [30], centered in $\vec{P}_o$ and having an asymmetric variance given by $\sigma_j^{\pm} = cA_j^{\pm}$, where $c$ is a common modulation coefficient. Besides, the distribution is truncated at 1. Finally, we numerically estimate $c$ by generating sets of points in the parameter space whose coordinates in each axis are obtained from the split normal distributions mentioned for an initial guess of $c$. Through an iterative process we search the value $c = c^*$ for which a 95% of the points generated in the parameters space, yield efficacy estimations verifying $L(\vec{P}) > 0.05$. Once we have found the optimal value of the scaling coefficient, the reported uncertainty of the $j$-th parameter corresponds to 95% CI given the distributions we have used.

## RESULTS

In order to find the sets of parameters yielding best estimates of BCG efficacies according to our models, we have performed a series of numerical optimization procedures seeking for likelihood maximization. First, we are interested in addressing whether a unique likelihood maximum exists across the parameter space of each model or whether, on the contrary, there exist multiple parameter combinations associated to comparable values of $L(\vec{P})$ close to the maximum. This is an important point to address, since the existence of different maxima in a model would indicate the inability of the model to univocally

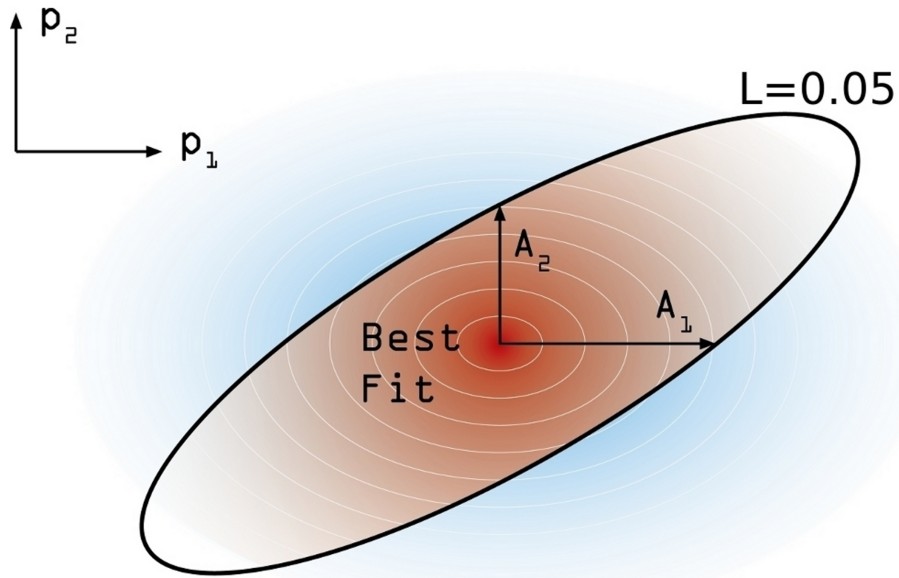

**Figure 4 Confidence intervals estimation scheme.** Degraded shades represent the joint probability density associated to the estimation of confidence intervals around the model best fit. The modulation coefficient $c$ is determined so as to make the brown area within the black line of $L(\vec{P}) = 0.05$ to precisely accumulate the 95% of the total joint probability distribution.

quantify the effects causing the efficacy variations observed. To tackle this question, we performed an iterative hill-climbing algorithm starting from 20,000 random points across the parameter space for each model. As it can be seen in Fig. 5, while model 2 presents a unique likelihood maximum ($L(\vec{P}^*) = 0.53$), models 1 and 3, which contemplates masking, fails at providing a univocal vaccine's description associated to a unique solution from likelihood optimization.

Instead of that, as we can see in Figs. 5A, 5C, 5D and 5E models 1 and 3 present a *parameters cliff* across which, model's likelihood is near to its maximum, and largely comparable ($L(\vec{P}^*) = 0.79$ for model 1, and $L(\vec{P}^*) = 0.002$ for model 2). Furthermore, a relative likelihood test comparing models 2 and 3 (that is, comparing blocking vs. masking as exclusive mechanisms) yields a relative likelihood $L_3(\vec{P}^*)/L_2(\vec{P}^*) = 0.002/0.53 = 3.8 \cdot 10^{-3}$. This result, considering that both models share the same amount of parameters, highlights again the inability of masking to provide a picture for vaccine efficacy variation as accurate as blocking does, as we can also see in Fig. 1, where the best fit provided by each model is presented as well.

If the analysis of model 3 and its comparison against model 2 allows us to discard masking as an autonomous mechanism able to explain the vaccine efficacy measured in the trials, it remains to be elucidated whether its consideration in addition blocking in model 1 might still be able to significantly improve the fitting of the observed data. To answer this question, we conduct a simple likelihood ratio test in which the null and full models are, respectively models 2 and 1. From such test, we obtain that the statistic: $\chi^2 = -2\ln(L_2(\vec{P}^*)/L_1(\vec{P}^*)) = 0.80$, is a chi-square distributed variable with 2 degrees

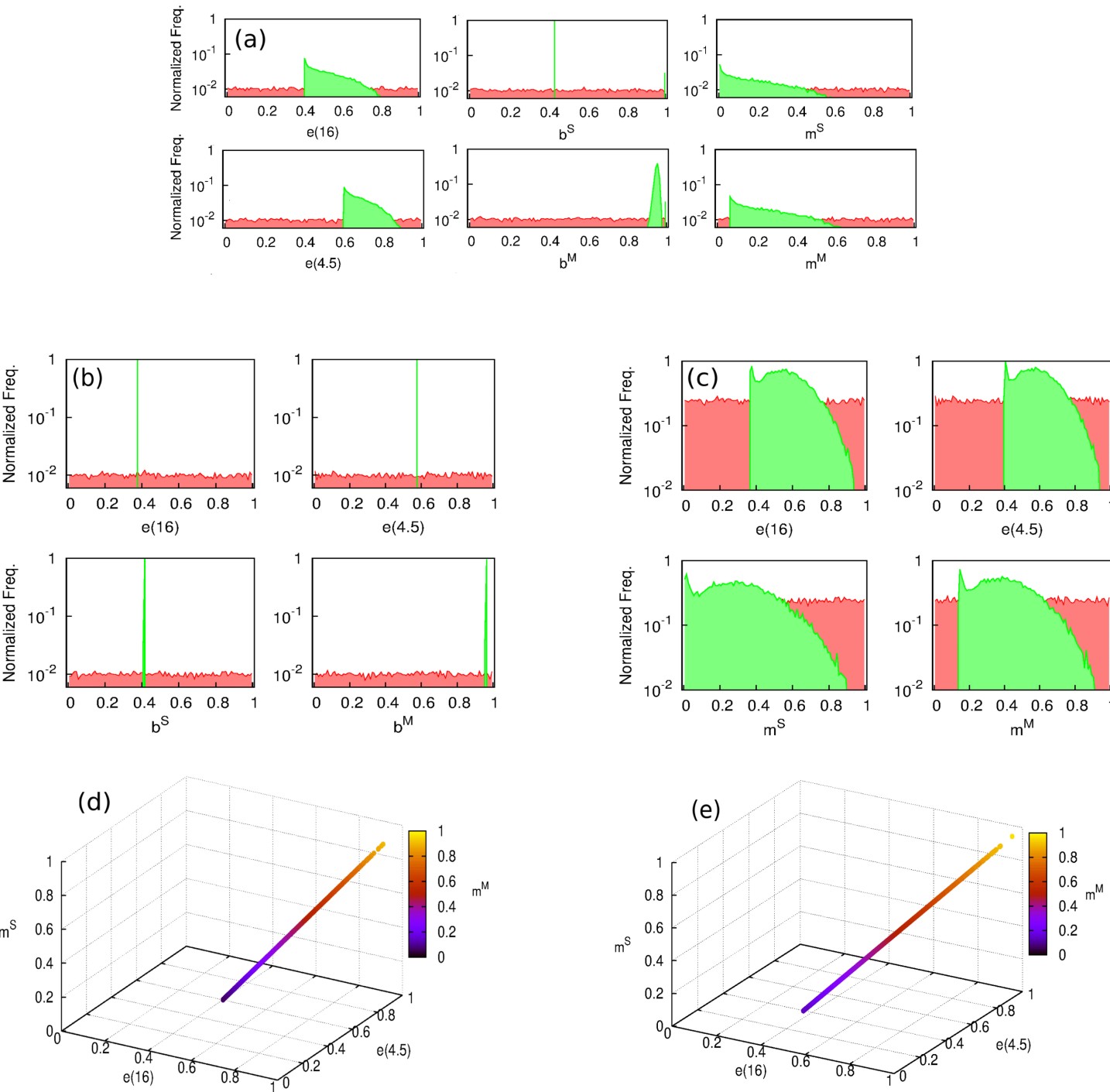

**Figure 5  Distribution of the parameters which yield a maximum in the likelihood function.** (A–C) Hill climbing algorithm distributions for models 1, 2 and 3, respectively. Starting from a series of randomly distributed points in the parameter space (their coordinates distributions are represented in red), a random displacement following a uniform distribution in the parameter space within a hyper-cube of size $d = 0.001$ is attempted at each time step, and accepted only if it corresponds to an increasing of the likelihood function $L(\vec{P})$. The algorithm stops when no further move is accepted after $N = 10^7$ rejected displacements (i.e., the function $L(\vec{P})$ reaches a maximum). 
**Figure 5 (…continued)**
In green, we see the peaked distribution of the end points of the algorithm around the solution of the models. (D and E) parameters cliff yielding quasi-constant values of maximum likelihood $L(\vec{P}*) = 0.79$ for model 1 and $L(\vec{P}*) = 0.002$ for model 3. As it can be seen in (A) and (D), the model versions that contemplate masking are unable to provide a clear univocal vaccine description yielding maximum likelihood. The reason for this behavior is the existence of a region in the parameters space, represented in (D) and (E), within which, likelihood is almost constant and close to its absolute maximum.

**Table 1  Optimal parameters of model 2.** Models 1 and 3 are unable to provide a unique parameter set yielding maximum likelihood.

| Parameter | Model 2 (only blocking) |
|---|---|
| $e(4.5)$ | 57.7% (46.8%–68.6%) |
| $e(16)$ | 37.6% (29.3%–45.8%) |
| $b^M$ | 96.4% (51.9%–99.8%) |
| $b^S$ | 41.1% (14.2%–68.0%) |

of freedom (difference between number of parameters of models 1 and 2) under the null model. The obtained value does not allow to discard it even with a 50% confidence ($\chi^2(p = 0.5, df = 2) = 1.39$), which indicates that masking is not just unable to provide an acceptable description of the observed data by itself but also makes no significant contribution to explain the variations in vaccine efficacies observed in the trials under study, when considered in addition to blocking. This is also reflected in the close estimates that are found for blocking parameters in models 1 and 2 (see Table 1 and Fig. 5).

Besides, if we analyze the combination of parameters that formed the cliff of maximum likelihood in model 1, we see that it consists in very similar levels of masking for the two different cities, which enters into conflict with the mentioned correlation between ES effects and closeness to equator.

In summary, our results point at blocking as the only plausible source of vaccine efficacy variation between the two mechanisms considered, validating the qualitative interpretation of the BCG-REVAC outcomes by *Barreto et al. (2014)*. The best fit of model 2 yields a likelihood $L(\vec{P}*) = 0.53$, which corresponds to moderate blocking levels in Salvador ($b^S = 0.41$ c.i. [0.14, 0.68]) and to almost total blocking in Manaus ($b^S = 0.96$ c.i. [0.52, 1.00]). These results are consistent with the assumed correlation between ES action strength and closeness to equator.

## DISCUSSION

Understanding the mechanisms driving ES effects on BCG performance is a crucial task in the agenda towards the development of new tuberculosis vaccines. In this work, we have proposed a mathematical model that allows the quantitative evaluation of these two effects based on the BCG-REVAC trials performed in Brazil (*Barreto et al., 2014*). We have seen that the divergence in the measured efficacies of the trials is explained with high values of blocking, which concur with the qualitative discussion made in *Barreto et al. (2014)*. Furthermore, we have also observed for the first time that no alternative behavior of BCG is compatible with the observed data within the context of a model in which BCG's variability is entirely attributed to ES sensitization.

Admittedly, the range of applications of the results here exposed must be restricted to the provision of a plausible explanation for the efficacy variation patterns observed within a controlled context such as the BCG-REVAC studies, in which the equivalent design of the trials in both cities makes reasonable to assume that ES is the only mechanism responsible for the variations observed. In that sense, the quantitative conclusions reached in this work should be interpreted as a mean to discriminate between the two mechanisms studied when it is reasonable to neglect any source of vaccine variation foreign to ES. Nonetheless, it is worth remarking that our approach cannot provide any insight on the relevance of ES itself when compared to other plausible sources of variability that could also affect vaccine's efficacy, such as diversity in production, administration, and type of BCG vaccine strain used, as well as the TB strain that circulates in a particular population, among others.

In that sense, the analysis of new, hypothetical trials similarly structured, conducted in other geographical areas, could certainly yield different results, if additional sources of variation not considered in this work were playing a relevant role.

The model here proposed could however be generalized so as to address some additional questions that go beyond a simple comparison between masking and blocking, which involve a more detailed description of the blocking effect itself.

On one hand, it is pertinent to ask whether prior vaccination with BCG might trigger a blocking effect comparable in magnitude to that caused by ES; a question that could be tackled by an extension of the model here proposed in which two blocking parameters-one associated to each source-are considered instead of one. Remarkably enough, distinguishing between BCG vaccine and ES as possibly different causes of blocking might lead to relevant quantitative consequences in what regards impact evaluation of novel vaccines, mostly because the old vaccine is still used in the vast majority of the countries, also in geographical areas in which low levels of ES would be expected.

On the other hand, an additional limitation of our study, inherited from the BCG-REVAC studies design, is due to the restriction of trials' endpoints to diseased and not diseased individuals, without measuring infection as a third relevant outcome. This limitation prevents us to address the important question of whether the vaccine is blocked in its protective role against infection, or if, instead, blocking interferes more intensely with the vaccine's performance at reducing the progression rates from latency to active disease (*Soysal et al., 2005*; *Roy et al., 2014*).

If infection was registered as an additional endpoint in the clinical trials—something, in general, feasible (*Andrews et al., 2015*)—our approach could then be extended so as to estimate two different blocking components associated to the impairment of vaccine's protection against infection and active disease independently. Once again, such hypothetical study could bring important insights for future vaccine development, and in particular could contribute strongly to the debate of what should be the primary goal of TB vaccines (*Hawn et al., 2014*). Generally speaking, more studies are needed to evaluate how general are the patterns found by BCG-REVAC trials, with the ultimate goal of assessing a positive explanation to the long lasting problem of BCG efficacy variation patterns.

TRIAL DESIGN → **clinical trial** → VACCINE EFFICACY → **spreading model** → VACCINE IMPACT

**Figure 6** Scheme of the basis for evaluation of anti tuberculosis vaccines in absence of universally reliable protection correlates. First stage: design of vaccine efficacy determination clinical trials: the age of the cohorts must be elected taking into account that prior exposure to mycobacteria—either environmental, *M. tuberculosis* after exposure or even prior TST or also BCG—may corrupt the observed vaccine efficacies. Second stage: vaccine impact evaluations: bulk, short-term and long-term impact forecasts should be equally considered, as well as age-distributed impacts in terms of cases, infections and casualties prevented.

## CONCLUSIONS

The crucial implications of discriminating and quantifying masking and blocking effects for TB vaccine development are twofold. On one hand, understanding the range, and causes behind variations of BCG efficacy is essential (*McShane, 2014*), since the efficacy of any novel vaccine will be measured against BCG. On the other hand, depending on where a new vaccine is applied and how old are the target populations, masking or—more likely—blocking effects would affect new vaccines too.

These issues affect different stages of the vaccine development pipeline, as sketched in Fig. 6. In the first place, during the process of vaccine evaluation in the context of clinical trials, studies of new tuberculosis vaccines should account for the possibility that prior sensitization may compromise their effects (*Mangtani et al., 2014*). In this sense, and even if a new vaccine targeting TB in adolescents and adults rather than any other age group is expected to have the quickest impact on disease transmission and control, before we address the question of impact of novel vaccines, it is essential to know if the vaccine is more effective than BCG. The most reliable way of knowing whether a new vaccine works better than BCG is by conducting an efficacy trial in a naive population without previous ES (e.g., previous BCG vaccination, mycobacterial infection and/or TB contact) in order to avoid possible effects of masking or blocking (*Rodrigues, Diwan & Wheeler, 1993*; *Andersen & Doherty, 2005*; *Barreto et al., 2014*).

Furthermore, and once the efficacy estimation is complete, in order to produce any reliable vaccine impact and cost-effectiveness forecast, modeling scenarios contemplating ES deleterious effects on TB vaccines are mandatory. The fact that, according to our analysis, blocking emerges as the driving effect behind BCG variability poses a potential pitfall to any vaccination strategy focused on individuals older than those analyzed here, including most strategies conceived so far for booster vaccines. This is especially worth noticing because blocking, unlike masking, is not supposed to degrade the vaccine-induced protection obtained further during life by individuals immunized promptly after birth. Again, even if immunizing adolescents is thought to provide better impacts than vaccination strategies focused on younger age-segments, if such a novel vaccine is affected by blocking just as BCG is, then its impact will decrease in a way that, given the high blocking levels here identified, might even revert the comparison. As suggested by Helen McShane "we should

optimize deployment of BCG to administration as close to birth as possible" (*McShane, 2014*). This should be the case for new priming live vaccines candidates based on BCG replacement strategies as well (*Marinova et al., 2013*).

Taken all together, our results highlight the need for measuring ES effects on novel vaccines performance, as well as of diversifying vaccination strategies.

### Funding

SA was supported by the FPI program of the Government of Aragón, Spain. JS was supported by the program of Postdoctoral Scholarships for Excellence of the Sainte-Justine UHC Foundation and by the Merit scholarship program for foreign students (PBEEE) of the Fonds de Recherche of Quebec, Nature et Tecnologies (FRQNT). This work has been partially supported by "Gobierno de Aragón/Fondo Social Europeo" and MINECO through Grant FIS2011-25167 to YM BIO2014-52580P, TBVAC2020 (643381) funded by the European Commission Horizon 2020 CM and DM; and the European FP7 grant NEWTBVAC 241745. Comunidad de Aragón (Spain) through FENOL to YM; and the EC Proactive project MULTIPLEX (contract no. 317532) to YM. The funders had no role in study design, data collection and analysis, decision to publish, or preparation of the manuscript.

### Grant Disclosures

The following grant information was disclosed by the authors:
FPI program of the Government of Aragón.
Postdoctoral Scholarships for Excellence of the Sainte-Justine UHC Foundation.
Merit scholarship program for foreign students (PBEEE), fonds de recherche Nature et Technologies Quebec.
Gobierno de Aragón/Fondo Social Europeo.
MINECO: FIS2011-25167, BIO2014-52580P.
European Commission Horizon 2020: TBVAC2020 (643381).
European FP7: NEWTBVAC 241745.
Comunidad de Aragón (Spain): FENOL.
EC Proactive project MULTIPLEX: 317532.

### Competing Interests

CM is a co-inventor on a composition of matter patent: Title: Tuberculosis Vaccine, Entidad Titular (Owner entity): Universidad de Zaragoza, N de Solicitud (Request number): PCT/ES 2007/070051.

### Author Contributions

- Sergio Arregui and Joaquín Sanz conceived and designed the experiments, performed the experiments, analyzed the data, contributed reagents/materials/analysis tools, wrote the paper, prepared figures and/or tables, reviewed drafts of the paper.
- Dessislava Marinova, Carlos Martín and Yamir Moreno analyzed the data, wrote the paper, reviewed drafts of the paper.

## Data Availability

The research in this article did not generate any raw data.

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
