# Peer review of "On the impact of masking and blocking hypotheses for measuring the efficacy of new tuberculosis vaccines"

_PeerJ, doi:10.7717/peerj.1513_

## Round 0.1 · original submission · Minor Revisions

You have comments from two expert referees, one of whom has chosen to waive anonymity. Both reviewers have made modest suggestions for changes and both feel it is publishable if properly revised.

Reviewer 1 ·

Basic reporting

No Comments

Experimental design

No Comments

Validity of the findings

No Comments

Additional comments

The manuscript entitled ‘On the impact of masking and blocking hypotheses for measuring efficacy of new tuberculosis vaccines’, by Arregui et al. addresses an important point looking at the correlation between the environment in blocking or masking the BCG protection against the development of tuberculosis (TB). This study is based on a Brazilian vaccination multi-study. The mathematical model proposed is well rationalized, as well as are the results and conclusions addressing the limitations of the presented model.

Just some minor comments depicted below to excel this study:
It is important that authors provide a stronger rational for the chosen study, as well as discuss other studies related to the impact of the environment in BCG protection. In this matter, Fig. 6 is quite simplistic addressing the impact of environmental factors. In this context, for example, there are not conclusive results about the impact of parasitic infections or non-tuberculous mycobacteria against the protective role of BCG in the development of active TB (either by the masking or blocking hypothesis).

In general terms, authors could also discuss (and no generalize) that variations in clinical trials may also be due to other important factors, including (but not limited to) the diversity in production, administration, and type of BCG vaccine strain used.
Also authors could pronounce about the controversy of looking at BCG protection against infection vs. protection about the development of the disease, and the implications of this in their model.

·

Basic reporting

Although the authors do an excellent job of motivating this study and presenting the importance of the specific question regarding the blocking and/or masking of BCG effects, they would strongly benefit from further editing for English language usage. Some of their word choices make it challenging to comprehend what are intrinsically difficult concepts (for example: distinguishing masking and blocking from blocking alone) and this detracts somewhat from what is otherwise a model of a clearly stated scientific problem. Figures 2 and 3 are extremely helpful but I wonder if Figure 3 might be improved by the addition of the expected protection in the non-vaccinated cohorts as well so that the reader can be reminded that VE is the comparison at time t of these two groups. I do realize that might make the figure overwhelmingly complex.

I also wonder if it might not be helpful to present some more detail on the trial design earlier in the paper. I was very relieved when I finally came to the detailed explanation of the design since I had not really understood what was being compared prior to that despite multiple references to the study.

Experimental design

The authors have done an outstanding job in defining a clear and important research question and should be congratulated for framing so precisely a series of hypotheses around the variable effects of masking and blocking that previously eluded this kind of clarity.

Validity of the findings

no comments.

Additional comments

I very much appreciated the rigorous and clear thinking behind this study. While I realize that it is impossible to capture all the biological possibilities that masking and blocking might encompass in this kind of model, I worry that some of the assumptions, while reasonable, are speculative and should be more strongly identified as such in the discussion. Specifically, it is not clear to me that masking would always be complete, ie. that it would bring vaccine efficacy back to its original value. Further, I think its possible that blocking works differently for BCG and other ES so it might not be reasonable to assume a single b parameter for both effects. I am not suggesting further exploration of these possibilities here, only more emphasis on possible limitations in the discussion.

---

## Round 0.2 · accepted · Accept

The referees from the first round have delegated the decision to me, and I am satisfied that your revisions have addressed their original concerns.